# Geohelminths: Use in the Treatment of Selected Human Diseases

**DOI:** 10.3390/pathogens13080703

**Published:** 2024-08-20

**Authors:** Magdalena Szuba, Weronika Stachera, Adrianna Piwko, Marianna Misiak, Renata Rutkevich, Marcin Sota, Lana Atrushi, Leyla Bennacer, Deborah Nzekea, Yen Ching Wu, Arya Taesung Kim, Subin Yu, Nash Ribeiro, Monika Dybicz

**Affiliations:** Department of General Biology and Parasitology, Medical University of Warsaw, 02-004 Warsaw, Poland; s082741@student.wum.edu.pl (M.S.); s082714@student.wum.edu.pl (W.S.); s074243@student.wum.edu.pl (A.P.); s087165@student.wum.edu.pl (M.M.); s087106@student.wum.edu.pl (R.R.); s087180@student.wum.edu.pl (M.S.); s082922@student.wum.edu.pl (L.A.); s083840@student.wum.edu.pl (L.B.); s087239@student.wum.edu.pl (D.N.); s080830@student.wum.edu.pl (Y.C.W.); s079366@student.wum.edu.pl (A.T.K.); s079381@student.wum.edu.pl (S.Y.); s079353@student.wum.edu.pl (N.R.)

**Keywords:** Crohn’s disease, ulcerative colitis, multiple sclerosis, allergic rhinitis, asthma, celiac disease, *Trichuris suis*, *Necator americanus*

## Abstract

Research on the therapeutic use of parasites has been ongoing since the development of the “hygiene hypothesis”. Parasites can stimulate the Th2-dependent response and suppress the Th1-dependent response, which is intensified in many diseases, especially allergic and autoinflammatory ones. In this review, we present the types of parasites used in helminth therapy and the range of diseases in which they may be useful. We also present the results of clinical trials conducted so far, which confirm the safety of such therapy and provide promising outcomes.

## 1. Introduction

Parasites are still a significant threat to human health and life. Despite this, research on the therapeutic use of parasites is being conducted around the world. Over 30 years ago, D. P. Strachan concluded in his work that allergic diseases, which are becoming more and more common in the population, are associated with a decrease in the number of infectious diseases [1]. The author observed that hay fever and eczema were less common in children from large families, who were probably exposed to more infectious agents than children from small families. This was revolutionary because it was previously widely believed that the increased occurrence of allergies was the result of increased pollution. This statement of Strachan was the basis for the so-called “hygiene hypothesis”. Over time, it was also extended to autoimmune diseases [2]. It has also been shown that a previous parasitic infection reduces the risk of allergies [3]. A continuation of this issue is the “old friends hypothesis” proposed by Graham Rook in 2003 [4]. According to it, the important factors are not infections, but rather microbes, such as viruses and worms, colonizing mammals and humans during the evolution. In 2010, Matricardi introduced the “microbial diversity” hypothesis [5]. It was developed a year later by von Hertzen [6]. Scientists have assumed that the key factor in regulating the immune system is the diversity of microbes in the intestines. The “biome depletion” theory is an extension of the above statements [7]. It assumes that the depletion of the human biome results from the use of toilets and water treatment plants, as this interrupts the life cycle of most worms. Effective drugs have been developed for helminths whose cycle do not depend on the above-mentioned sanitary solutions. Therefore, currently humans are the only vertebrate in which these parasites are not commonly found. The remedy for allergic and autoinflammatory diseases may be to recreate the ecosystem of the human body using helminths. It is worth mentioning Dr. David Pritchard from the University of Nottingham, who conducted research in Papua New Guinea in the late 1980s. He noticed that Papuans infected with the hookworm *Necator americanus*, which lives in the human intestines, did not suffer much from a number of autoimmune diseases. In 2004, the scientist infected himself with *N. americanus*, in order to develop a theory explaining this phenomenon. After this experiment, Dr. Pritchard received permission to conduct research involving volunteers [8].

In this review, we present several potential parasitic candidates and their possible therapeutic use in particular diseases or groups of diseases. Currently, scientists mainly consider two species of parasites transmitted through the soil, so-called geohelminths: *Trichuris suis* and *Necator americanus*. Both of these species are representatives of worms, so they fit the “old friends hypothesis”. They may be useful in the treatment of inflammatory bowel disease, allergic diseases (allergic rhinitis, asthma) and multiple sclerosis.

## 2. Inflammatory Bowel Disease

Inflammatory bowel disease (IBD) is a common name that covers two diseases: Crohn’s disease and ulcerative colitis. These are chronic, inflammatory diseases of the gastrointestinal tract with unknown causes [9].

### 2.1. Crohn’s Disease

Crohn’s disease has a complex aetiology, including genetic susceptibility, environmental factors and abnormal intestinal microflora. All this causes an abnormal immune response of the gastrointestinal mucosa. The disease appears in the form of relapses and remissions. The typical patient is young, presents with right lower quadrant abdominal pain, chronic diarrhoea and weight loss. Extraintestinal skin, joint or ocular symptoms may precede diagnosis by up to 50% [10]. In Crohn’s disease, the intestinal mucosa is inflamed, mainly by Th1 and Th17 cells [11]. Researchers also report abnormal function of Treg cells in this disease [12]. Normally, Treg cells regulate the activity of Th1 and Th17 and prevent the spread of inflammation. In Crohn’s disease, the inflammatory activity of Th1 and Th17 cells is greater than the ability of Tregs to modulate them what results in overactivity of Th1 pathways. Colonization with helminths enhances the Th2 response by stimulating the production of IL-4 and IL-13, what in turn inhibits the Th1 response [13].

#### 2.1.1. *Trichuris suis* in Crohn’s Disease Therapy

It has been proposed that Crohn’s disease is common in parts of the world where helminth colonization is rare but absent in populations where parasite carriage is common [14]. One of them is *T. suis*. It is a parasite typically found in pigs [15]. It has a direct life cycle in which there is no intermediate host (Figure 1). Infected animals excrete single-cell, initially non-infectious eggs in their faeces. The egg develops into the infectious L1 larva within 3 weeks to 2 months. It is very resistant to environmental conditions, so it can survive in the soil for several years. When the infectious L1 egg is ingested by another animal, the larva hatches in the small intestine and cecum. Over the next 5 weeks, the larva undergoes 4 moults (L2, L3, L4) until the adult stage (L5) [16]. Females reach 6–8 cm, and males reach 3–4 cm. Most individuals are located in the cecum and the initial part of the colon. The lifespan of *T. suis* is 4–5 months. *T. suis* is closely related to *T. trichiura*, but in most humans, it causes only self-limiting colonization [17]. There are suggestions that most *T. suis* larvae after hatching in the human gastrointestinal tract remain immature and live there for several weeks. However, *T. suis* can sometimes be invasive in humans, being able to mature to adult size and reproduce in humans [18].

Japanese researchers decided to evaluate the safety of using *T. suis* in humans. 12 healthy volunteers were given single doses of *T. suis* ova (TSO) in the amount of 1000, 2500 or 7500 eggs. Observation lasted 56 days. During this time, 3 study participants experienced moderate symptoms, such as loss of appetite, flatulence and diarrhoea. None of them required hospitalization or antiparasitic treatment. This randomized, double-blind, placebo-controlled study suggests that the use of TSO seems to be safe [19]. Other researchers have proven that adult individuals of *T. suis* produce excretory/secretory proteins (TsESP) that inhibit the production of pro-inflammatory cytokines, but also induce the secretion of IL-10. TsESPs also induce nitric oxide production and arginase-1 expression and inhibit CD4+ T cell proliferation [20]. Another study investigated the effect of *T. suis* excreted/secreted products on epithelial barrier function. These products reduce epithelial function through glycan-dependent downregulation of claudin-4 and tight junction proteins EMP-1. Perhaps in the future these glycans will be isolated and can be directly administered to patients, without the need for infection with live parasites [21].

In 2005 Summers et al. conducted a 24-week clinical trial to assess the safety and effectiveness of *T. suis* therapy in Crohn’s disease (Table 1) [22]. The study included 29 patients with moderate symptoms. Each patient received 2500 live TSO every 3 weeks. 80% of patients responded to treatment and 73% of patients experienced remission. Unfortunately, a large placebo effect cannot be ruled out because it was an open study. It is worth adding that *T. suis’s* treatment did not cause any complications or side effects. It was noted that patients using immunological therapy at the same time had better treatment results. Similar observations concerned patients in whom the disease process spared the terminal ileum. 12 years later, Schölmerich et al. conducted the first randomized, double-blind, placebo-controlled study assessing the effectiveness and safety of using TSO in the treatment of Crohn’s disease. 252 patients with mild and moderately advanced disease were included in the 12-week study. Patients treated with immunosuppressive or biological drugs were excluded from the study. Patients in the study group received TSO every 2 weeks in one of 3 doses: 250, 2500 or 7500 eggs. None of these doses showed clinically significant induction of remission compared to placebo. It is important that administration of even large doses of TSO (7500) was safe, showed a dose-dependent immune response and did not cause serious side effects (Table 1) [23].

#### 2.1.2. *Necator americanus* in Crohn’s Disease Therapy

*N. americanus*, also known as New World hookworm, is an intestinal parasite of humans and another parasite useful in the treatment of Crohn’s disease. It can be found in Africa, Asia, Australia and the Americas, making it cosmopolitan, although with preference for warm and moist climates, where larvae can survive in the environment. *N. americanus* embryos hatch in the soil, where they moult twice, developing into the infective third-stage filariform larva (L3) (Figure 2). Upon contact with a potential host, L3 larva actively penetrates the skin in order to reach either the blood or lymphatic vessels. Upon entering the vasculature, it migrates to the lungs, passing to alveoli and finally bronchioles. From that point, larva is dependent on the host’s autonomic reflex. It requires to be coughed up in order to reach the pharynx and then swallowed, to pass to the gastrointestinal tract, where it will reside in the small intestine in its mature form [24]. *N. americanus* infections are mostly asymptomatic. The presence of parasites, however, can be indicated by skin-related inconveniences or pulmonary and gastrointestinal signs in the early phases of necatoriasis. L3 larvae can cause transient symptoms such as urticarial dermal reaction at the site of penetration, or mild pneumonia, sore throat, cough, red sputum, and headache during their migration.

Early intestinal infection can be associated with nausea, abdominal pain, enteritis, and weight loss. It is worth noting, however, that juvenile hookworms are the ones responsible for the damage to intestinal mucosa. The tissues have shown to be remarkably altered during the endoscopic investigation conducted in the presence of mature worms [25]. Seldom does necatoriasis cause iron deficiency anaemia or cardiac complications. Fortunately, the infection can be easily managed via a three-day treatment with 100 mg of oral Mebendazole taken twice a day [26].

For the sake of experimental infections with *N. americanus*, inoculations with L3 larvae are performed. The site of choice is most commonly the skin of an arm, which presents with vesicles at each invasion point, especially in cases of doses exceeding 10 filariform larvae [25]. Previously, it was postulated that the therapeutic effect is dependent on the level of *N. americanus* invasion, which was determined by establishing the minimum number of eggs in one gram of stool. However, it has been proven that the increase in the number of eosinophils in peripheral blood as a result of infection with this helminth occurs independently (Table 2) [27]. Experimental groups consisting of infected individuals have shown increased incidence of skin rash, pain events, and flatulence in comparison to control groups receiving placebo. These symptoms, however, steadily subsided over the course of the trial. Surprisingly enough, when analysing the well-being of both groups of the 2011 trial, the experimental group showcased less lethargy than the control (Table 2) [28]. Taking all the aforementioned factors into consideration, the surveyed infections were proven to be safe and were generally well-tolerated by the trial participants.

Kabeerdoss et al. conducted a case-control study examining whether prior hookworm infection provided any protection against Crohn’s disease in Indian patients in Vellore (Table 2) [29]. It tested the hypothesis that Crohn’s disease should exhibit an inverse relation with exposure to hookworm. The hypothesis was explored with the intention of only looking at an association between the two and was not an attempt at finding an explanation between the association. A venous blood sample of among 78 Crohn’s disease patients and 75 healthy controls was taken, and lymphocyte proliferation in response to hookworm antigens in vitro was used as a marker of prior hookworm infection. T cell activation was measured by calculating the mean (SD) change in CD3+CD69+ and CD3+CD45RO+ cells upon introduction of crude extracts of hookworm antigens. To measure cytokine activity, ELISPOT was used to assess interferon-g response to a panel of 6 recombinant hookworm antigens. Participants were also controlled for potentially confounding factors, such as socioeconomic status and residence. Patients with Crohn’s disease had a significantly lower shift in CD3+CD69+ cell population and less interferon-g response than the controls. CD69 antigen is expressed on T cells soon after exposure to infectious antigens and the lower change in their expression indicate a reduced exposure to hookworms in Crohn’s disease patients. On the other hand, interferon-g response represents a hookworm infection that has been cleared after therapy. CD3+CD45RO+ cells did not exhibit notable changes in any groups. There is probably a protective mechanism against Crohn’s disease following hookworm infection.

### 2.2. Ulcerative Colitis

Ulcerative colitis is an another disease in IBD. It is idiopathic and, unlike Crohn’s disease, affects only the mucous membrane. The disease process usually begins in the rectum and develops proximally, affecting part or all of the colon. The disease progresses with periods of exacerbations (bloody diarrhoea, abdominal pain) and remission [30,31]. Ulcerative colitis is characterized by an atypical Th2 response. It is mediated by “natural killer” T cells, which produce IL-5 and IL-13, which are cytotoxic to epithelial cells [32].

#### *T. suis* in Ulcerative Colitis Therapy

Summers et al. conducted a study analogous to that on Crohn’s disease, but including patients with ulcerative colitis. This time the study was randomized, double-blind and placebo-controlled. The 12-week study included 54 patients with active ulcerative colitis. Members of the study group received 2500 TSO every 2 weeks. Statistical analysis suggests that patients with completely colonic involvement and shorter disease duration responded better to *T. suis* therapy. It should be noted that the conclusions are limited due to the small sample size. Clinical improvement was observed in most patients, but few disease remissions occurred. No treatment-related side effects or complications were observed (Table 1) [33]. In 2024, Prosberg et al. conducted a randomized, double-blind, placebo-controlled clinical trial. 60 patients with moderately active ulcerative colitis took 7500 TSO every two weeks for 24 weeks. TSO treatment induced a temporary remission at week 12, but no better clinical remission was achieved at week 24 compared to placebo. The most frequently reported adverse events were: diarrhoea, upper abdominal pain, and flatulence. Moreover, 4 patients from the study group required colectomy during TSO therapy [34].

## 3. Celiac Disease

Celiac disease (CeD) is an autoimmune condition affecting the gastrointestinal tract, triggered by gluten intolerance. Clinical manifestations typically include gastrointestinal symptoms such as diarrhoea, malabsorption, and weight loss, commonly they are linked to consuming gluten-rich grains such as wheat, barley, and rye. While some individuals primarily experience gastrointestinal symptoms, others may exhibit extraintestinal manifestations or be diagnosed through family history screening, even in the absence of gastrointestinal complaints [35]. The hallmark of the majority of patients with CeD is the presence of specific autoantibodies (anti-tissue transglutaminase (TG) and anti-endomysium (EMA) antibodies). 

### N. americanus in Celiac Disease Therapy

The double-blinded, placebo-controlled clinical trial assessed the impact of *N. americanus* infection on the immunotoxic effects of gluten in otherwise healthy patients with CeD, with an evaluation of tolerability of the infection [28]. From weeks 0 to 20, 20 participants maintained a strict gluten-free diet. Inoculation with 10 3rd-stage hookworm larvae or placebo and with 5 infective 3rd-stage hookworm larvae or placebo was performed at week 0 and week 12 of the trial, respectively. No serious adverse reactions caused by the *N. americanus* inoculations were detected with haemoglobin levels not being affected. A slight leukocytosis and eosinophilia were detected from week 4 post-*N. americanus* infection. Expectedly, IFN-γ-producing T cells were detected after the wheat challenge but there was no significant difference in frequency of those cells between the groups. No apparent effect of *N. americanus* inoculation on the disease was detected (Table 2). The histological damage observed was contradictory to the immunological responses, as it occurred in the participants of the trial following gluten consumption, regardless of the presence of hookworm in their bodies. Such results could have been caused by the low quantities of *N. americanus* administered and the steep reintroduction of gluten into the diet. It is believed that innate inflammatory responses, such as intraepithelial lymphocyte migration and enterocyte proliferation, might have masked any effect on adaptive immunity, which is the characteristic feature of CeD (Table 2). 

Following these studies, 20 patients with celiac disease on a long-term gluten-free diet were infected percutaneously with 10 L3 larvae or given placebo [36]. After 12 weeks, a dose of 5 L3 larvae (or placebo) was administered. At week 20 post-infection, all subjects were subject to a gluten challenge consisting of four slices of white bread per day for 5 days. In second trial seven participants were infected with *N. americanus*, boosted and challenged with gluten in an identical manner to that described for first trial. As previously showed, hookworm larvae administration used in celiac disease did not significantly suppressed pathology. However reduced mucosal inflammation was observed after hookworm infection and gluten challenge. Another trial ongoing 52-week involved 12 patients with CeD [37]. They were inoculated with 20 3rd-stage larvae and challenged gluten diet (pasta administered). The gluten challenges revealed decreased IFNγ and increased Treg cells. The results demonstrated that treatment of CeD patients using *N. americanus* larvae infection and gluten microchallenge promoted tolerance and in addition can stabilize the indices of gluten toxicity.

A randomised, double blinded, placebo controlled clinical trial was also run for 94 weeks on 54 patients with celiac disease using third stage *N. americanus* larvae (Table 2) [38]. At the start of the trial the two groups L3-20 and L3-40 received 2 doses of hookworm of ×10 and ×20 respectively 8 weeks apart. The gluten challenge was carried out in 5 phases to test different levels of tolerance to gluten. During these phases blood, faecal and duodenal biopsy were collected. Most common adverse events seen at 12 weeks were diarrhoea, nausea, fatigue, etc. and during the gluten challenge phase (weeks 12–42) the adverse events were similar to the first 12 weeks. Noteworthy, is that the hookworm group displayed significantly fewer adverse events than the placebo group during the gluten challenge phase. No participant was seen to have gotten the major clinical effect of hookworm infection, anaemia. There were 7 out of the 54 participants that withdrew from the study. Twenty two out of the remaining 47 participants completed the study successfully with Marsh 0 or 1 grade histology score and normal tTG (transglutaminase antibody—0–3 U/mL); 4 out of 7 in placebo group, 14 out of 32 in L3-20 group and 4 out of 8 in L3-40 group. Anti-tTg IgA antibody level was stable until week 36 when it rose significantly at phase 4 (sustained gluten) but relative change was similar between the 3 groups. Celiac Disease Quality of Life Measure (CD-QoL) score significantly improved with no difference between the 3 groups. These results show that there was not a definite protection against an anti-gluten free diet using hookworm infection.

## 4. Diabetes Mellitus

Type 2 diabetes is a metabolic condition to affecting hyperglycaemia and insulin production leading to resistance. A double randomised, double-blinded, placebo-controlled clinical trial was run for 2 years in Australia on type 2 diabetes people to determine the effect of third stage *N. americanus* larvae infection on the insulin resistance (Table 2) [39]. The primary goal of the study was the safety and tolerability of hookworm treatment. 13 patients received placebo treatment, 14 participants received a dose of 20 hookworm larvae (L3-20) and 13 received a dose of 40 hookworm larvae (L3-40). There were a total of 20 adverse events but none were serious. One GI related syndrome was severe,3 (1 from L3-20, 2 from L3-40 group) were removed early and given deworming medications. Two recovered swiftly and 1 person persisted. No GI related adverse events were noticed in the placebo group. No clinical effect, anaemia, related to moderate to heavy hookworm infection were noticed. There was a significant number of dropouts throughout the trial, with 24 of the 40 randomised participants completing the trial: 8 (62%) from placebo group, 7 (50%) from L3-20 group and 9 (69%) form L3-40 group. Diet and exercise habits that could affect metabolic health were collected through regular questionnaires throughout the trial. Fasting blood glucose values showed: stability in placebo group, in both L3-20 and L3-40 groups fasting this parameter reduced from 5.2 mmol/L to 4.5 mmol/L and 5.3 mmol/L to 4.3 mmol/L respectively at 6 months of the trail. At 12, 18, and 24 months there was a significant decrease in the values in the L3-40 group. Compared to the placebo treatment there was no significant difference in the L3-20 group but a significant difference was present at 6 and 12 months in the L3-40 group. In placebo group insulin levels were stable at 6 months but HbA1c showed an increase reaching maximum of 35 mmol/L after 2 years. In L3-20 group insulin levels were lower to 10 mU/L from 13 mU/L at 12 months but HbA1c levels were stable. In group L3-40 there was no change in insulin level but HbA1c levels had a similar increase like the placebo group at 18 months. The values of body mass and BMI in the placebo group were stable. In L3-40 no consequential changes were seen. However, in L3-20 the observed values were lower than baseline, especially at 18 and 24 months Participants that were infected with low doses of hookworm showered major reduction in glucose homoeostasis compared to the placebo group. Although the study was small the trial was able to confirm safe dose and its associated improvements in glucose homoeostasis in people with risk of type 2 diabetes.

## 5. Allergic Diseases

Allergy is a civilization disease of the 21st century. In the United States and Europe, allergic rhinitis affects between 20 and 30 per cent of adults [40]. This disease cause sleep disturbances, decrease work productivity and negatively affects social interactions. Patients with moderate or severe allergic rhinitis have a higher risk of anxiety, depression and fatigue compared to patients with mild allergic rhinitis [41]. There is no cure, but many medications are available to relieve symptoms, and desensitisation to allergens is sometimes used. However, new treatments are constantly being sought because symptomatic treatment is not effective enough. There is an interest in understanding how parasite infection affects allergen hypersensitivity and inflammation. Bager and al. conducted a randomised, double-blind, placebo-controlled clinical trial to investigate the effect of TSO in allergic rhinitis (Table 1) [42]. The study included 100 people aged between 18 and 65 years suffering from grass pollen allergy. Some of them took a total of 8 doses of 2500 live TSO, while the others took a placebo. Treatment with TSO resulted in transient diarrhoea, increased eosinophil counts and *T. suis* specific IgE, IgG, IgG4 and IgA compared to placebo. There were no significant changes in runny nose, itching, sneezing, total histamine, grass-specific IgE, skin prick testing to grass and 9 other allergens. The results showed that the *T. suis* group had a significantly higher rate of moderate to severe gastrointestinal symptoms that occurred consistently. There was no significant difference between treatment groups in the rate of mild gastrointestinal symptoms [42].

A very similar trial was conducted and published in 2012 [43]. One hundred adults with symptomatic allergic rhinitis and grass pollen allergy were randomised into two treatment groups. The first group was given a suspension of 2500 TSO and the second group was given a placebo liquid at intervals of 21 days over a period of 6 months. Blood samples were taken from each patient before the start of the study, during the study and 21 days after the end of TSO or placebo administration. The severity of allergy symptoms and the levels of plasma cytokines were assessed. Helminth infection induced a Th2 response with elevated levels of IL-5 as well as parasite-specific IL-4, IL-5 and IL-13. It also increased levels of the regulatory cytokine IL-10. TSO treatment had no effect on allergen-specific cytokine responses and the global profile of peripheral blood mononuclear cell cytokine responses during the grass pollen season [43]. Efficacy of *T. suis* in relieving rhinitis has not been proven in trials. Not only therapeutic successes are important but also how the treatment affects the body. 

## 6. Asthma

Hookworm infections have been linked to beneficiary effects regarding the respiratory system. In particular they have been associated with the reduction of wheezing and prevention of asthma symptoms (Table 2) [44,45]. Since allergic diseases are less common in rural areas where the rate of parasitic infections is higher than in urban areas known for high rates of allergic infections and almost no prevalence of parasites. It has been hypothesized that helminths by interacting with immunological network might play a protective role against various allergens. Hookworm infections are associated with interaction with regulatory T-cell and anti-inflammatory IL-10 that are believed to be a link between parasitic infection and their effect on allergies [46]. A clinical trial of experimental infection with *N. americanus* that primary measured provocation dose of inhaled adenosine monophosphate (AMP) required to reduce forced expiratory volume in 1 s by 20% (PD20AMP) showed that after 16 weeks from the cutaneous administration of either ten *N. americanus* larvae, or histamine solution (placebo) there were improvements in both groups. After 16 weeks PD20AMP improved in 69% of participants who received hookworm and 50% in those who received placebo. The differences between groups were not statically significant. In addition, self-reported asthma symptom questionnaires improved in both groups, even slightly more in the placebo group than in the hookworm group; the answers also did not differ significantly between groups. The only statically significant difference between the study and the control group was the presence of hookworm infection symptoms such as localized skin itching and redness, nausea and abdominal pain [27]. Moreover, a dose-ranging study conducted showed that there was no change in functioning of the lungs after being subjected to *N. americanus larvae* [47]. In both studies the infection with *N. americanus* was well tolerated, and led to increased eosinophil count in the group infected with hookworm (Table 2) [27,47].

## 7. Multiple Sclerosis

Multiple sclerosis (MS) is a chronic inflammatory central nervous system disease. There are hypotheses that the high incidence of MS may be due to the increase in hygiene today, including reduced exposure to parasites. Parasites have been shown to affect the immune system through various immunomodulatory mechanisms. The promising results of studies on the use of TSO in therapy in patients with inflammatory bowel disease suggest that the immunomodulatory effect of therapy could also be achieved in MS. For this reason, a study was conducted using *T. suis*, which immunomodulatory effects were thought to affect the course of the disease in people with MS. Both the safety of TSO therapy and its efficacy were investigated. In each study, patients were orally administered 2500 TSO in a regimen of every 2 weeks, and the duration of therapy was several months. All conducted studies [48,49,50,51] (in which study groups consisted of 10 to 12 patients) concluded the safety of oral use of TSO. The treatment was well tolerated, with the only side effects in patients being mild gastrointestinal symptoms (no interference with daily activities such as school or work, 2–3 loose stools per day), which were self-limiting and lasted from a few days to a few weeks. Blood parameters such as differential blood count, kidney (creatinine, urea) and liver (GGT, ALAT, ASAT, AP) parameters or electrolytes were all within the reference range [51]. As mentioned helminths exhibit a variety of immunomodulatory effects. Studies [49,50,51] have shown that they have the effect of increasing the production of anti-inflammatory cytokines such as IL-4, IL-5, IL-10 and IL-13 what leads to induction of a Th2 response, decrease in Th1 response and changes in the total number of CD4+ and CD8+ cells (decreases in CD4+ and CD8+ T cells). A frequently noted change in patients treated with *T. suis* was eosinophilia (Table 1) [49,50,51], which appeared in more than half of the patients. The levels of eosinophils initially increased during the first two months of TSO administration, and then decreased in most patients during and after the last month of exposure to TSO. It is probable that these trends reflect an early inflammatory response at the onset of gastrointestinal infection, followed by a later anti-inflammatory response induced by sustained low-level colonization of *T. suis* and repeated oral dosing of TSO. Besides blood tests, patients were monitored by magnetic resonance imaging (MRI). In one study [50], the average number of new MS-typical lesions enhancing after gadolinium administration in a masked reading of double-dose gadolinium administration studies was 6.6 at the beginning of screening, 2.0 at the end of 3 months of TSO treatment and 5.8 at 2 months after the end of treatment. In another study [51] of 12 MS patients with parasitic infection over a follow-up period of 4.6 years, the patients showed significantly lower disease exacerbation and reduced MRI changes as compared to uninfected MS patients. In the other observational study [49] in a small group of patients with relapsing-remitting MS, oral *T. suis* therapy was well tolerated but without beneficial effects. To summarize the results of the above studies, longer studies with larger numbers of participants will be needed to determine whether these results represent disease regression, early therapeutic response or coincidence, and will properly determine the safety and efficacy of TSO in the treatment or adjunctive treatment of MS.

**Table 1 pathogens-13-00703-t001:** *T. suis* therapy in human clinical trials.

Disease	Dose of Parasite	Number of Patients	Method	Result	Safety Concerns	Reference	Ref No
Crohn’s Disease	2500 live TSO every three weeks for 24 weeks	29	Open-label study	79.3% responded and 72.4% achieved remission at week 24. Mean CDAI decrease of 177.1 points.	No adverse events reported. Therapy was well-tolerated even among those receiving multiple immunosuppressants.	Summers et al., 2005	[22]
Crohn’s Disease	250, 2500, or 7500 TSO every two weeks for 12 weeks	252	Randomized, double-blind, placebo-controlled trial	No significant advantage over placebo in inducing remission or clinical response.	Safe but no serious adverse reactions specifically attributed to TSO.	Schölmerich et al., 2016	[23]
Ulcerative Colitis	2500 TSO every 2 weeks	54	Randomized controlled trial	Clinical improvement in most patients; few achieved remission.	No serious side effects reported, but full safety profile requires further research.	Summers et al., 2005	[33]
Ulcerative Colitis	7500 TSO every 2 weeks for 24 weeks	60	Randomized, double-blind, placebo-controlled clinical trial	No better clinical remission was achieved at week 24 compared to placebo.	4 patients required colectomy during TSO therapy.	Prosberg et al., 2024	[34]
Allergic Rhinitis	2500 TSO every 21 days, 8 doses	100	Randomized, placebo-controlled, double-blind trial	No significant improvement in allergy symptoms; increased eosinophils and specific antibodies.	Increase in gastrointestinal symptoms noted; potential allergenicity not fully evaluated.	Bager et al., 2011	[42]
Allergic Rhinitis	2500 TSO every 21 days over 6 months	100	Randomized, placebo-controlled, double-blind trial	TSO induced a Th2-polarized response and elevated IL-5, without affecting allergen-specific cytokine responses.	Elevated IL-5 and parasite-specific cytokines; no alteration in allergen-specific reactivity during peak allergy symptoms.	Bourke et al., 2012	[43]
Multiple Sclerosis	2500 TSO orally every 2 weeks for 12 weeks	10	Open-label, magnetic resonance imaging assessor-blinded, baseline-to-treatment study	No significant beneficial effect observed; increase in MRI lesions and relapses during treatment.	Well-tolerated but associated with gastrointestinal symptoms and eosinophilia.	Voldsgaard et al., 2015	[49]
Multiple Sclerosis	2500 TSO orally every 2 weeks for 3 months	5	Phase 1 study	Reduced number of new gadolinium-enhancing MRI lesions. Preliminary increase in anti-inflammatory cytokines IL-4 and IL-10.	Well-tolerated with no significant adverse effects observed during the study duration.	Jo et al., 2011	[50]
Multiple Sclerosis	2500 TSO orally every 2 weeks for 6 months	4	Immune monitoring study	Slight downregulation of Th1-associated cytokine IL-2, temporary increase in Th2 cytokine IL-4. Mild eosinophilia and changes in T-cell and NK cell numbers observed.	Well-tolerated with mild eosinophilia. No significant safety concerns reported.	Benzel et al., 2011	[51]

The gut microbiome has been implicated to play an important role in natural immunity and maintaining the organism’s homeostasis. The studies have shown MS to be one of the autoimmune diseases displaying abnormalities in its α-diversity. According to analyses, recorded dysbiosis included reduced *Parabacteroides*, *Prevotella*, and *Bacteroides* (Bacteroidetes), *Adlercreutzia*, *Collinsella* and *Slackia* (Actinobacteria), Erysipelotrichaceae and Veillonellaceae families (Firmicutes) and Proteobacteria such as *Sutterella*, but also increased abundance of *Acinetobacter calcoaceticus*, Caulobacteraceae, *Pseudomonas*, *Mycoplana*, *Methanobrevibacter*, *Akkermansia*, *Haemophilus*, *Blautia,* and *Dorea* [52,53,54]. Immunological and microbial changes start taking place years before the onset of MS, hence making it difficult to establish whether dysbiosis is a cause or a result of the autoimmune disease. According to the study concerning the effects of necatoriasis in patients with MS published in 2021, the data showed the greatest differences between gut microbiota of infected and hookworm-free individuals 9 months after parasite inoculation. Bacterial taxa associated with the gut microbiota of relapsing MS patients, such as the Lachnospiraceae family, including the genera *Roseburia*, *Dorea*, and *Tyzzerella* were significantly expanded in the control group compared to infected individuals. Experimental group also displayed increased Mollicutes class, which is otherwise found to be reduced in the gut microbiota of the diseased cohort. Additionally, the study showcased the difference between microbial diversity of infection responders (N+responders) and non-responders (N+non-responders). *Parabacteroides* were found to be significantly more abundant in N+responders compared to N+non-responders in the stool samples taken a week before the beginning of the trial. The presence of the aforementioned genus was identified as the top ranking biomarker of treatment outcome via machine learning. Flavobacteriaceae family (Bacteroidetes), which was repeatedly reported to be depleted in people suffering from autoimmune conditions, was also proven to be consistently increased in the fecal microbiota of N+responders [55]. The overall conclusions arising from the article are favorable to the hypothesis of a contributory role of parasite-associated modulation of host bacterial microbiota composition, and encourage further studies on the topic.

**Table 2 pathogens-13-00703-t002:** *N. americanus* therapy in human clinical trials.

Disease	Dose of Parasite	Number of Patients	Method	Result	Safety Concerns	Reference	Ref No
Asthma	10 larvae	32	Randomized, placebo-controlled trial	Improvement in airway responsiveness was non-significant; well tolerated with mild symptoms.	Infection was well tolerated without significant exacerbation of asthma symptoms.	Feary et al., 2010	[27]
Celiac Disease	15 larvae (10 at week 0 and 5 at week 12)	20	Randomized double-blinded placebo-controlled trial	No improvement in primary outcomes; infection safe and tolerable. Experiment imposed no obvious benefit on pathology.	Infection well tolerated; some transient pain and enteritis. No serious adverse effects.	Daveson et al., 2011	[28]
Wheeze in Ethiopia	Not specified	604 (cases and controls)	Nested case-control study	Significant reduction in the risk of wheeze, suggesting a protective effect against asthma symptoms.	Well tolerated with no specific safety concerns.	Scrivener et al., 2001	[44]
Asthma	Not specified	Multiple studies included	Systematic review and meta-analysis	Significant reduction in asthma risk. The highest tertile of infection intensity showed the strongest protective effect.	Not specified in meta-analysis, but general trends suggest that hookworm infections are well tolerated.	Leonardi-Bee et al., 2006	[45]
Dose-ranging Study	10, 25, 50, 100 larvae	10	Dose-ranging study	All administered doses achieved the target of at least 50 eggs/gram of feces. Infection elicited a modest host eosinophil response and was suitable for use in preliminary clinical therapeutic trials.	Skin itching and gastrointestinal symptoms common at higher doses, but overall well tolerated with no significant change in lung function.	Mortimer et al., 2006	[47]
Celiac Disease	10 L3s at 0 and 12 weeks	Not specified	Controlled clinical trial	Modulation of immune response to gluten, enhancing mucosal IL-1β and IL-22 while suppressing IFNγ and IL-17A levels.	Safe with no serious adverse effects noted, well tolerated with mild symptoms. Further studies recommended.	Croese et al., 2013	[25]
Celiac Disease	20 or 40 larvae	54	Randomized, placebo-controlled trial	No restore tolerance to sustained moderate consumption of gluten (2 g/d) but improved symptom scores after intermittent consumption of lower gluten doses. Participants reported fewer gluten-related adverse events.	Well tolerated with no serious adverse effects; mild gastrointestinal symptoms were the most commonly reported issues.	Croese et al. 2020	[38]
Crohn’s Disease	Crude extracts of hookworm antigens	153 (cases and controls)	A case-control study	Hookworm antigen decreased immune reaction in patients with Crohn’s disease by lowering shift in CD3+CD69+ cell population and interferon-g response than the controls.	Hookworm crude extracts were generally well tolerated; some transient pain and enteritis. No serious adverse effects.	Kabeerdoss et al. 2011	[29]
Multiple Sclerosis	25 larvae transcutaneously	71	Double-blind, randomized, placebo-controlled trial	Fewer new lesions and relapses observed; significant decrease in disease exacerbation and MRI changes.	Generally well tolerated with mild symptoms; further studies needed to fully assess the safety profile in larger and more diverse populations.	Tănăsescu et al., 2020	[56]
Type 2 Diabetes (Insulin Resistance)	20, 40 larvae	40	Randomized, placebo-controlled trial	Significant improvement in fasting glucose and insulin resistance at 1-year	Infection was well tolerated with some gastrointestinal symptoms	Pierce et al., 2023	[39]

An article published in 2020 suggested a possible therapeutic effect of the microbiome in MS [56]. At the end of the study, 51% (n = 18/35) of MS patients experimentally infected with *N. americanus* showed no detectable new lesions, as assessed by MRI scans, in comparison to 28% (n = 10/36) of placebo-treated volunteers. There were also 5 relapses (14.3%) in the hookworm group and 11 (30.6%) in the control group. Indeed, the hookworm infection had proven to provide the hosts with an anti-inflammatory effect. Percentages of eosinophils and of CD4+CD25highCD127negT cells in peripheral blood of worm-colonized individuals were significantly increased 9 months post-infection compared to control group (Table 2).

## 8. Conclusions

The main candidates for use in helminth therapy have potential use in human diseases such as inflammatory bowel diseases, allergic diseases, celiac disease and multiple sclerosis. A number of studies have been conducted involving these parasites. The first clinical studies concerning helminth therapy in humans started after 2000. Over the past 2 decades, dozens of clinical trials have been conducted on humans. After 2000, only a few papers were published showing the potential of helminth therapy in human diseases.

The results were not spectacular, but it should be noted that they were performed on groups of patients with severe symptoms in whom all previous therapeutic lines had failed. It is worth adding that the presented studies confirmed the safety of helminth therapy, because infections were either asymptomatic or with mild symptoms. One of the most controversial aspects of helminth therapy is the need to induce infection in humans. Although no serious side effects were found in the above studies, it should be remembered that before settling in the intestine, helminth larvae migrate through various tissues of the body, where they can cause an inflammatory reaction. Therefore, it is worth conducting research to avoid the stage of human infection. It may be possible to do this using cell lines to multiply ex vivo populations of immune cells that appear in parasitic infection. Such actions would also affect the psychological aspect of helminth therapy. Administering only cells would also remove the mental barrier against undertaking such therapy on the part of both patients and doctors. In the future, it is worth to continue the work on these parasites, searching for their specific molecular features that may be useful in treatment, but also conducting multicenter, high-class clinical trials with patients in order to use the therapeutic potential of parasites, including species other than those mentioned.

## Figures and Tables

**Figure 1 pathogens-13-00703-f001:**
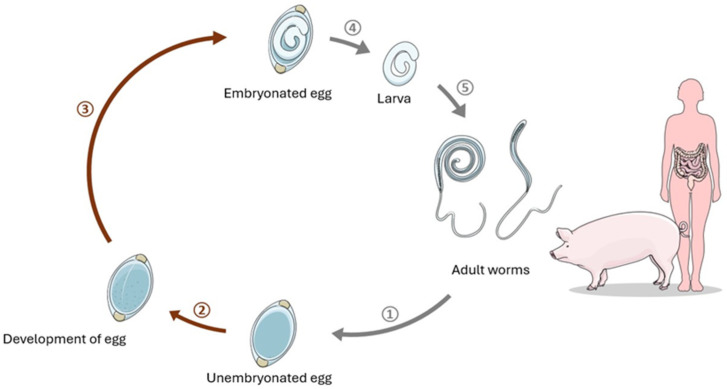
The life cycle of *T. suis.* 1. Fertilized adult worm lays unembryonated eggs, which are excreted in the faeces. 2. Unembryonated egg developes in soil. 3. Unembryonated egg becomes embryonated in warm and moist condition. Man acquires infection by ingestion of contaminated soil, food and water. Pigs are the natural host for *T. suis*. 4. Larva hatches through the pole of the egg in the small intestine. 5. Larva undergoes moults to the adult worm in the mucosal layers. The most of adult worms are located in the large intestine. There are suggestions that most *T. suis* larvae after hatching in the human gastrointestinal tract remain immature and live there for several weeks.

**Figure 2 pathogens-13-00703-f002:**
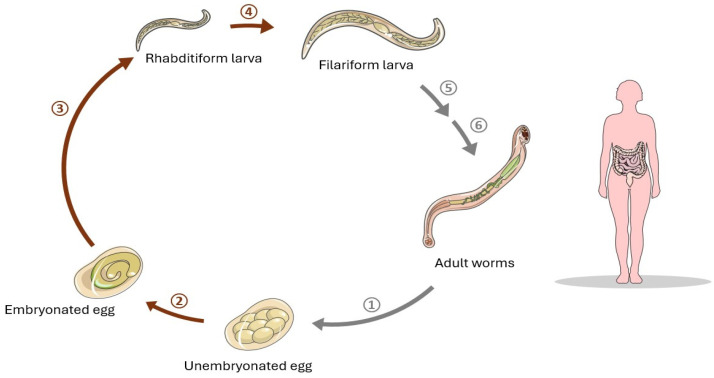
The life cycle of hookworm. 1. Fertilized adult worm lays unembryonated eggs, which are excreted in the faeces. 2. Embryonated egg is developed in moist, sandy and warm soil. 3. Rhabditiform larva hatches out from the egg. 4. Rhabditiform larva develops to filariform larva. 5. Humans become infected by penetration of skin (bare foot in dampen soil) by filariform larva. The larva is rarely transmitted via oral, transplacental or transmammary routes. 6. Larva migrates through the bloodstream to the lungs. In the lungs, the larva moves up the respiratory tract, is swallowed, and reaches the small intestine where the larva matures into adult hookworm. The adult hookworm attaches to the intestinal mucosa by their teeth in buccal capsule.

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
