# Peer review of "Geohelminths: Use in the Treatment of Selected Human Diseases"

_pathogens, 2024, doi:10.3390/pathogens13080703_

Round 1

Reviewer 1 Report

Comments and Suggestions for Authors

While I think helminth therapy is an interesting area of research which could benefit from being reviewed in the “pathogens” journal, I think there is very considerable scope to improve the manuscript that Szuba et al. have submitted and that publishing it in its current form would not provide a useful contribution to the field. Below are some ways I think the authors could revise their manuscript so that it might be turned into a useful contribution to the field and become an article that I would be willing to support being published in pathogens. I would gladly provide more detailed comments on a revised manuscript that has undergone a major re-write in-line with my suggestions.

Revise the title and abstract so they better reflect the content of the manuscript

The title is very broad and, like the abstract, does not reflect the content of the review and thus, also like the abstract, should be changed to reflect the actual content of the manuscript. The term helminth is used to for wide range of pathogenic infections including infections caused by filarial parasites and a broad range of geohelminths most of which are not discussed at all in the present manuscript. The current manuscript makes no mention of filarial parasites and is focused entirely on two species of geohelminths Trichuris suis (a zoonotic whip worm) and one hookworm species of Necator americanus.

Provide a clear narrative

The introduction is presently very discursive opening up with a discussion about malaria which is a subject the paper then does not return to. It does contain some interesting historical perspectives on the “hygiene hypothesis”, but needs to explain more clearly what has been done since the hypothesis was first propose and why it is timely to review the advances made since now. The introduction should also make very clear what the manuscript’s objectives are and how it plans to reach them. By the end of the introduction, it should be clear why the manuscript is so focused on Trichuris suis and Necator americanus and why there is so little about the other helminths. The manuscript’s subheadings should give the manuscript a clear direction that leads naturally to its conclusions. Presently the subheading don´t form a narrative thread and the current conclusion section reads like a second abstract. A conclusion section should not begin with “to summarize”. The conclusions should not be a summary (that is the job of the abstract) it should be a series of new insights brought together by assembling the latest research highlighted in the review.

Enrich the content by substantially updating and expanding the literature discussed and cited

The manuscript only cites 5 research papers from the last 5 years and none from the last 3 years and cites fewer than 50 papers in total. At present the manuscript reads like a good undergraduate essay rather than an expert review of the current state of a field. The review admirably focused on double-blind placebo-controlled clinical studies (which are the best source of data for the subject they are reviewing), but does little more than repeat the conclusions of the studies it refers to. The authors need to draw on many sources of research beyond these studies and weave them together to construct novel insights. More must be said about other helminthic infections and their effects on human pathology, immuno-modulation and allergic diseases. How can lab studies help us done with other helminth parasites help us to interpret the clinical trials they refer to?

1.      Remove repetition

There is a lot of repetition in the current manuscript. There is very significant overlap in the content included in the legends of figures 1 and 2 and the content of the manuscript´s text, which should be corrected. Similarly, there is currently a lot of repetition about the side effects observed from different TSO clinical studies as well as the immune response these treatments provoke. Restructuring the manuscript could help to avoid this. At present the manuscript first discusses Trichuris suis (a zoonotic whip worm) therapeutic studies and then Necator americanus studies afterwards. If the manuscript was restructured to first deal with the side effects of geohelminth therapies and then with the immune response to them and was then followed by a section on their therapeutic efficacy for each disease in turn (i.e having just one Crohn’s disease section)  a lot of repetition could be avoided. Discussing Trichuris suis and Necator americanus therapies together could also help the authors to assemble novel insights about geohelminth therapeutics in general. For example, they could comment generally on geohelminth therapeutics efficacy of Crohn’s  disease in general and the general immune response and side effects of these therapeutics.

Comments on the Quality of English Language

The english, in general, is fine. 

Reviewer 2 Report

Comments and Suggestions for Authors

Many reviews have already been written on helminth therapy and its immune and molecular mechanisms, often repeating the same information, including the associated diseases. Therefore, I see no originality or benefit in adding another review on this topic.

Moreover, I find it confusing that the authors include an introduction about Plasmodium, a blood protozoan, when the article's focus is on intestinal helminths. Additionally, the abstract and title suggest a comprehensive summary of therapeutic helminths, yet two significant species are missing from the review.

Key articles in the field of helminth therapy are also absent. In my opinion, this review fails to contribute anything new to the existing body of knowledge.

Round 2

Reviewer 2 Report

Comments and Suggestions for Authors

Recommend to include "biome depletion theory" that is most relevant nowadays than hygiene hypothesis and Old Friends one - of course both of these are very important. 

Authors have improved the article greatly! Congratulations.

Author Response

Comments and Suggestions for Authors

Recommend to include "biome depletion theory" that is most relevant nowadays than hygiene hypothesis and Old Friends one - of course both of these are very important. 

Authors have improved the article greatly! Congratulations.

Response:

Dear Reviewer,

Thank you for taking the time again to assess our manuscript. We appreciate your valuable recommendation to include "biome depletion theory". We added we a section containing this theory and the relevant literature on it.